# Transposon Insertion Mutagenesis in Mice for Modeling Human Cancers: Critical Insights Gained and New Opportunities

**DOI:** 10.3390/ijms21031172

**Published:** 2020-02-10

**Authors:** Pauline J. Beckmann, David A. Largaespada

**Affiliations:** 1Department of Pediatrics, University of Minnesota, Minneapolis, MN 55455, USA; pjjackso@umn.edu; 2Masonic Cancer Center, University of Minnesota, Minneapolis, MN 55455, USA; 3Department of Genetics, Cell Biology and Development, University of Minnesota, Minneapolis, MN 55455, USA; 4Center for Genome Engineering, University of Minnesota, Minneapolis, MN 55455, USA

**Keywords:** animal modeling, cancer, transposon screen

## Abstract

Transposon mutagenesis has been used to model many types of human cancer in mice, leading to the discovery of novel cancer genes and insights into the mechanism of tumorigenesis. For this review, we identified over twenty types of human cancer that have been modeled in the mouse using *Sleeping Beauty* and *piggyBac* transposon insertion mutagenesis. We examine several specific biological insights that have been gained and describe opportunities for continued research. Specifically, we review studies with a focus on understanding metastasis, therapy resistance, and tumor cell of origin. Additionally, we propose further uses of transposon-based models to identify rarely mutated driver genes across many cancers, understand additional mechanisms of drug resistance and metastasis, and define personalized therapies for cancer patients with obesity as a comorbidity.

## 1. Transposon Basics

Until the mid of 1900’s, DNA was widely considered to be a highly stable, orderly macromolecule neatly organized into chromosomes. Barbara McClintock challenged this paradigm in 1950 when she published her studies on the first transposable elements, *Ac* and *Ds*, which she discovered in maize [1]. She found that these transposable elements, or transposons, could cause large genetic changes and reversibly alter gene expression. Transposons have been classified based on their mechanism of movement throughout the genome (transposition). Class I is made up of retrotransposons which mobilize through an RNA intermediate-based “copy-and-paste” mechanism. This review will focus on class II elements, which use a DNA-mediated “cut-and-paste” mode of transposition. In nature, transposons encode an enzyme to direct their transposition called a transposase, and transposase recognition sequences on both ends (terminal inverted repeats (TIRs)), which direct transposase binding and mobilization of the transposon. For use in a laboratory setting, the transposon and transposase can be physically separated, with the transposase supplied in *trans*. This allows the transposon to encode alternative DNA sequences and for the system to be more intricately regulated. Transposon insertion is a mutagenic process and can result in both gain and loss of function mutations.

Transposition technology can be used in both “forward” and “reverse” genetic studies. Reverse genetics involves targeting a specific gene of interest to facilitate gain or loss of function studies. For example, knocking out or overexpressing a putative oncogene in a relevant cell line and analyzing the resultant phenotypic changes. These studies are quite useful for validation and functional analysis of single genes but are limited in their scope. Forward genetic studies obtain a phenotype through mutagenesis on a genome-wide scale, allowing the study of many genes and pathways simultaneously. For example, chemical mutagens, ionizing radiation, or transposition can be used to create a desired phenotype (i.e., change in leaf structure or tumor formation), and then mapping of the associated genetic changes will give insight into what genes or gene sets are involved in the phenotype under study.

Transposons have been used to study gene function successfully in many organisms, including yeast, plants, invertebrates, and vertebrates. For example, the prokaryotic bacteriophage *Mu* transposition complex has been used to disrupt gene expression in yeast, mouse, and human cells [2]. The maize DNA transposons *Ac/Ds*, *En/Spm*, and *Mu* have been used in maize, rice, tomato, and *Arabidopsis* [3,4,5,6,7]. The *Drosophila mauritiana* transposon *Mos1* has been used successfully in several forward genetic screens in *Caenorhabditis elegans* to identify important genes in a variety of biological processes [8,9,10,11]. *P* element transposons and transposable elements with diverse insertional specificities including *Tol2*, *piggyBac* (*PB*), and *Minos* have been instrumental to our current understanding of the *Drosophila melanogaster* genome [12,13,14,15]. *Tol2* (isolated from medaka fish) and insect-derived *PB* and *Minos* have also been used in mutagenesis in vertebrates such as the mouse and zebrafish [16,17,18]. *Sleeping Beauty* (*SB*) is derived from elements cloned from salmonid fish and has been widely used in insertional mutagenesis screens in mice [19,20,21,22,23,24,25,26,27,28,29] and shown to be active in other vertebrates including cultured cell lines, rats, zebrafish, and *Xenopus* [19,30,31,32].

The main practical differences between transposable elements include cargo capacity, integration site preference, and the rate of “local hopping.” Cargo capacity varies greatly among transposable elements; this is an important factor to consider, particularly for delivery of complex genetic cargos or longer genes. Transposition frequency of Tc1/*mariner* family members, including *SB* and *Minos*, decreases with increasing transposon length [33,34,35], although *SB* has shown to be able to deliver very large BAC constructs (>60 kb) [36] and has been modified to handle large sequences with more efficiency (>10 kb) [37]. *PB* and *Tol2* are more tolerant of increasing transposon size, making them a preferred choice for larger sequences [16,38]. Integration site preference is also important to consider when choosing the appropriate transposon vector. For use in mutagenesis, it is preferable to use a transposon system with a propensity to land within genes, like *PB*, to increase the chance of changing gene expression [39]. On the other hand, a nearly random mutagenesis system is likely to have less bias for a subset of genes. For use in a gene therapy setting, systems without a proclivity for transcriptional units like *SB*, are superior [40]. Some transposons display a sequence preference for integration, with Tc1/*mariner* elements (*SB*, *Frog Prince*, *Minos*, and *Hsmar1*) integrating into a TA dinucleotide sequence and *PB* targeting a “TTAA” sequence. In the case of *SB*, DNA structure and bendability are the primary predictive factor for integration and compared to other transposon systems, *SB* integration is affected little by gene content or other genomic features, making it an ideal tool for random mutagenesis [41]. Finally, local hopping, or a preference for transposons to land into cis-linked sites in close proximity of the donor locus, plays a significant role in the saturation efficiency during a mutagenesis experiment. *PB* and *SB* both exhibit local hopping, although *PB*-mediated local hopping is less pronounced [39,42]. Local hopping may be advantageous for a particular experiment, for example if saturation of a specific chromosome is of interest. If not, it can be circumvented by use of multiple transposon locations and/or taken into account during the analysis of the mutation data generated.

In comparison to other methods of identifying genetic drivers of cancer such as CRISPR/Cas9 or retroviruses, transposon insertion mutagenesis has its advantages and disadvantages (Table 1). While both CRISPR/Cas9 and transposon systems can cover the entire genome, transposon screens carry a slight bias related to local hopping and insertion preference that can be eliminated with careful guide RNA library design. However, in the context of in vivo models of cancer, CRISPR/Cas9 is hardly comparable to the utility of transposon mutagenesis. While CRISPR/Cas9 can be used to create loss and gain of function mutations, genome wide screens are done in such a way that each cell suffers a single mutation. Transposon mutagenesis in vivo allows for the accumulation of multiple, independent mutations that can cooperate to cause a phenotype. Therefore, transposon mutagenesis more accurately reflects the complexity of human cancer, which evolves in a stepwise manner. More recently these technologies have been combined by using a transposase (either PB or SB) to deliver single guide RNAs (sgRNAs) and Cas9 into mice in a reverse genetic approach [43,44]. Weber et al. delivered SB transposase and a transposon containing many sgRNA and *Cas9* sequences flanked by SB recognition sequences by tail vein injection resulting in the formation of hepatocellular carcinoma and intrahepatic cholangiocarcinoma [43]. This combination allowed delivery of multiple sgRNAs simultaneously and more high-throughput screening. Slow transforming retroviruses have been used to identify important drivers of mouse lymphoma (MuLV) and mammary tumors (MMTV) [45,46], however the application of these viruses is limited due to their cellular tropisms. The main advantage of transposon-based mutagenesis systems to retroviral screens is their tissue flexibility and the modifiable nature of the components, allowing tumorigenesis in nonlymphoid and non-mammary tissues.

## 2. Transposons to Model Human Cancer in Mice

Transposase systems, mainly *SB*, have been used to model and identify genetic drivers in many types of human cancer (Table 2). For use in forward genetic screens, the *SB* transposon and transposase have been modified to achieve sufficient mutagenesis to drive tumor formation (Figure 1A). The first transposons used, *T2/Onc* and *T2/Onc2*, use the murine stem cell virus long terminal repeat (MSCV-LTR) promoter followed by a splice donor (SD) sequence to drive gene expression and bidirectional splice acceptors (SA) and polyadenylation signal (pA) to terminate gene transcription and ablate expression [20,21]. This allows *SB*-mediated transposon insertion mutagenesis to identify both oncogene and tumor suppressor gene candidates (Figure 1B). An optimized *SB* transposase sequence (*SB11*) was knocked into the *Gt(ROSA)26Sor* locus, facilitating ubiquitous expression [21,34]. By crossing the *R26-SB11* mouse with mice carrying either *T2/Onc* or *T2/Onc2*, researchers were able to induce leukemia in mice (Figure 1C) [22]. Subsequently, a conditional *SB* mouse was created (*R26-lsl-SB11*), allowing tissue and temporal-specific transposition and modeling of very specific cancers [23]. For example, we used *Nestin*-driven *Cre recombinase* to drive *SB* expression solely in the developing central nervous system and to identify novel genetic drivers of childhood brain tumors [28]. The expression profiles of many of the Cre strains described in Table 1 have been characterized by The Jackson Laboratory [47]. While transposon-mediated mutagenesis screens have taught us a great deal about cancer development over the last two decades, we would like to focus on a few studies and overall lessons learned.

## 3. Cell of Origin

*SB* mutagenesis has been used to test the impact of the cell of origin and stage of differentiation on transformation potential. Berquam-Vrieze et al. initiated transposition at increasingly differentiated stages in T-cell development using Cre-inducible *SB* and 3 different *Cre* transgenes [78]. *Vav*-*iCre*, *Lck-Cre*, and *CD4-Cre* induce Cre expression in hematopoietic stem cells, immature T-cells without CD4 or CD8 expression, or late-stage T-cells expressing both CD4 and CD8, respectively. The authors found that *Vav-iCre* mice had a significantly shorter survival time, indicating hematopoietic stem cells are a more permissive cell for leukemia induction than more differentiated populations. In agreement with this, they found that there was an increased average number of driver insertion mutations per leukemia clone with increased differentiation. In other words, it took more “genetic hits” to transform a more differentiated cell of origin. This concept that transformation potential is lost with differentiation has been shown in other models, including intestinal cancer and medulloblastoma [107,108]. When Berquam-Vrieze et al. compared genetic drivers in tumors generated with the three Cre transgenes, they found significantly different gene profiles for each differentiation stage, suggesting that the biology of each cell of origin greatly affects the genetics of tumor development. Interestingly, Berquam-Vrieze and colleagues compared subsets of *SB*-induced mouse lymphoma and found that the *CD4-Cre* (most differentiated cell of origin) lymphoma matched the expression patterns of human ETP-ALL, a subtype of T-ALL defined by expression patterns of early T-cell precursors. This was unexpected, as this was the most differentiated cell of origin in the study. Therefore, this study sheds light on the potential cell of origin for human ETP-ALL, suggesting it may be a more differentiated T cell that regains expression patterns of earlier T-cell progenitors, rather than an undifferentiated more stem-like cell.

## 4. Identification of Rare Events

One challenge in human cancer genetics has been identifying rarely mutated driver mutations, including both tumor suppressors and proto-oncogenes. This has sometimes been referred to as the “long tail” problem, reflecting the large number of genes, that are altered in a relatively small percentage of cancer cases. Many such genes exist in a “gray area” and it cannot easily be determined if their alteration is selected for in human cancer development. Sequencing of many human cancer cases will be required to determine if their alteration is statistically significant [109]. Transposon-based forward genetic screens can provide contributing circumstantial data that such candidates may be driver alterations in cancer. Each screen that is completed reports a list of a few frequently mutated candidate genes and many more infrequently mutated candidates. When multiple screens are combined and analyzed together, infrequently altered drivers become more visible across many cancers. For example, *Rreb1* is a tumor suppressor gene that has been identified as a candidate driver in a low number of many tumor types, including intestinal and pancreatic cancer and B-cell lymphoma [24,59,85]. Another example is *Foxr2*. Transposon mutagenesis studies identified *Foxr2* as a strong candidate driver of malignant peripheral nerve sheath tumors (MPNST), osteosarcoma, and medulloblastoma [28,29,48,96]. Interestingly, human *FOXR2* is amplified and overexpressed in a subset of human MPNST and activated by translocation or amplification in a subset of human embryonal tumors of the central nervous system [29,110]. Various other studies indicate that *FOXR2* high level expression is a feature of a subset of many tumor types, where it is likely a driver mechanism [111,112,113]. 

Based on the nature of how transposon-based screens work, they are uniquely able to identify proto-oncogenes activated by creation of fusion transcripts. We queried the Candidate Cancer Gene Database of *SB* screen-derived cancer gene candidates for those with recurrent fusion transcripts among the TCGA [114]. Indeed, many of the *SB*-predicted oncogenes are activated by translocations similar to those described for *FOXR2*. This includes known proto-oncogenes like *ERG* and *RAF1*, but many more novel proto-oncogene candidates including *AMBRA1* and *RALY*. *AMBRA1* is a regulator of autophagy and has been shown to affect drug resistance in several cancers [115,116,117]. *RALY* is an RNA-binding protein implicated in metastasis and associated with poor prognosis in breast and colorectal cancer and hepatocellular carcinoma [118,119,120]. More analysis to identify novel fusion transcripts at the RNA level could identify more novel, poorly understood drivers that have been missed through traditional analysis but may have meaningful implications.

Thus, transposon-based screens pooled together can be used to identify and prioritize novel human oncogenes activated in a rare subset of many cancers. This is relevant for new clinical trial designs, called “basket trials,” in which a single drug is tested in a variety of tumor types with a specific genetic alteration. For example, chromosomal fusion events involving the carboxy-terminal kinase domain of *TRK* (tropomysosin receptor kinase) have been identified in many cancers and shown to drive constitutively active, ligand-independent signaling which results in tumorigenesis regardless of tissue origin [121,122,123,124,125,126]. Larotrectinib is a potent and selective inhibitor of TRK proteins and has shown a durable antitumor effect in patients with *TRK* fusions regardless of patient age or tumor type [127].

## 5. Drivers of Metastasis

Two screens, one in medulloblastoma and one in osteosarcoma, have sought to address the questions of clonality and drivers of metastasis [48,99]. These studies found that when drivers of primary and metastatic tumors were compared, there was varying degrees of overlap. In the case of osteosarcoma, there were rare instances where multiple metastatic tumors within the same mouse were significantly different from each other [48]. In the case of medulloblastoma, dissemination from the primary tumor likely occurred early in the tumor development, potentially in multiple “seeding” events or in an on-going fashion [99]. These findings indicate that metastatic cancer develops in a rare subclone, perhaps early after tumor development, and that using targeted therapies based on the drivers in the primary tumor will not be enough to eliminate metastatic tumors as other driver alterations may have taken over a primary role in tumor maintenance. Additionally, several strong candidate drivers of metastasis were identified in these papers, including alterations in *Pten*, *Gsk3b*, *Snap23,* and *Raf1* in osteosarcoma [48]. Loss of *PTEN* has since been identified as a marker of poor clinical prognosis and lung metastasis in osteosarcoma [128].

We believe that new transposon-based screens could be designed to better facilitate identification of metastatic drivers (Figure 2A). For practical purposes, a small screen can be done to produce a small number of primary tumors in mice expressing transposase, harboring a mutagenic transposon array, and any predisposing background mutations of interest. Primary tumors can then be removed and transplanted as allografts into multiple recipient mice. Ideally the tumors would be implanted orthotopically, and the primary tumor would be removed at a pre-defined size, allowing the metastases to expand. The timing of primary tumor removal will need to be optimized for every cancer type and experimental condition to balance death caused by the primary tumor and leaving the tumor in long enough for metastasis to occur. The metastatic clones can then be harvested and their genetic drivers identified in a much more expedited fashion compared to undergoing a full screen. The drivers identified in metastases can be compared to each other and the primary tumor to identify genes involved in metastasis and to provide knowledge on the clonality of the metastases in the cancer being studied. It may also be possible to provide adjuvant or neoadjuvant chemotherapy to better mimic the selective pressures that human metastases have undergone upon disease recurrence.

## 6. Therapy Resistance

Transposon-based mutagenesis in the presence of a targeted therapy offers a powerful tool for understanding genetic pathways to therapy resistance in cancer, which is a major problem in the quest for durable cures. For example, the *BRAF*^V600E^ mutation is present in approximately half of human melanoma, resulting in hyperactivation of the MAPK pathway [129]. While targeted therapy using vemurafenib was initially promising, tumors eventually recurred showing re-activation of the MAPK pathway [130]. Mann et al. performed *SB* mutagenesis in a *BRAF^V600E^*-driven mouse melanoma and identified many candidate cooperating genetic alterations [75]. Using a similar screening strategy, but including vemurafenib treatment of one cohort, Perna et al. were able to compare drivers in vemurafenib-resistant and treatment-naïve tumors, and these authors identified novel mediators of vemurafenib-resistance including *Eras* [74]. *ERAS* is an activator of PI3K/AKT signaling, which the authors show fosters resistance to vemurafenib through inactivation of the pro-apoptotic protein BAD. Therefore, dual treatment with a PI3K inhibitor along with vemurafenib may be a promising treatment in the clinic if observed therapy-related toxicities can be overcome (NCT01512251). 

In another example, Kas et al. studied resistance to AZD4547, a selective FGFR inhibitor, by orthotopically implanting an *SB*-accelerated mammary tumor with *FGFR2* activation into syngeneic FVB mice and treating with AZD4547 [54]. FGFR is upstream of both the MAPK-ERK and PI3K-AKT pathways and is frequently hyperactivated in human cancers [131,132]. Clinical trials of several FGFR inhibitors have shown success in a subset of patients, but mechanisms of FGFR-inhibitor resistance are still being understood [133]. Treatment resistant versus naïve tumors were compared by RNA-sequencing and analysis of transposon insertion mutations. The authors identified a diverse spectrum of resistance mechanisms to FGFR inhibition. Reactivation of the MAPK-ERK pathway was the dominant form of resistance, suggesting that combining FGFR and MEK/ERK pathway inhibitors may be the most effective strategy for patients with FGFR activation. In addition, the authors found that *Abcg2*, a drug efflux pump, expression was upregulated in some AZD4547 treated tumors, while inactivation of *Rasa1* was found in other AZD4547 resistant tumors. These results provide guidance that future drug design of FGFR inhibitors should be specifically made to be poor substrates for drug efflux pumps. More pre-clinical models of *SB*-mediated accelerated tumor evolution could be used to predict drug resistance mechanisms in the clinic to give additional options to patients and facilitate more intelligent drug design (Figure 2B). Similarly, to the metastasis experiments described above, these experiments could be done using tumors derived from a small screen allografted into several recipient mice. Such studies would ideally be done in immunoproficient mice in the context of recurrent metastatic disease, to best approximate the clinical situation for patients. Drivers in therapy-resistant tumors would then be compared to untreated tumor drivers to find pathways involved in drug resistance.

In addition, a similar in vivo screen has been used to gain insight into resistance to the MET inhibitor Fortinib in a model of *SB*-accelerated medulloblastoma [98]. The *PB* system was used to identify mechanisms of resistance to an Mdm2 inhibitor in *PB*-accelerated tumors of various types passaged as allografts [134]. Although the following studies were not carried out in vivo, but rather in cell lines, it is worth noting that transposon mutagenesis has been successfully used to screen for cancer cell drug resistance in several reports [134,135]. Taken together, these studies suggest that cancer evolution in response to the selective pressures of therapy can be usefully explored using transposon mutagenesis.

## 7. Obesity and Tumor Development

Worldwide, the incidence of obesity has nearly tripled since 1975 [136]. Of particular concern, the prevalence of overweight and obese children ages 5-19 has risen from just 4% in 1975 to over 18% in 2016 [136]. The fundamental cause for this increase in obese and overweight people is an increase in energy-rich food intake and a reduction in activity. Excess adipose tissue predisposes individuals to develop type 2 diabetes mellitus, cardiovascular disease, and several types of cancer [137,138,139]. Obesity has been associated with increased cancer risk in colorectal, kidney, pancreatic, gallbladder, thyroid, breast, ovarian, esophageal, liver, and endometrial cancer [140,141,142]. Increased BMI (body mass index) has been associated with reduced cancer survival and increased recurrence after radio- or chemo-therapy [143,144,145].

However, despite strong clinical, preclinical, and epidemiological evidence linking obesity to increased cancer risk [138,139,146,147], the mechanisms behind this are still not completely understood. Local dysregulation in adipose tissues of obese individuals results in systemic metabolic changes including insulin resistance, chronic inflammation, and hyperglycemia [148,149]. Dysregulated paracrine signaling from adipocytes shapes a permissive microenvironment to tumor development and progression through secretion of signaling molecules (including proinflammatory cytokines, proangiogenic factors, and adipokines) and by acting as an energy reservoir [139,150,151,152]. For example, chronic inflammation brought on by obesity results in increased expression of signal transducer and activator of transcription 3 (STAT3) and nuclear factor-κB (NF-κB), which increase cellular proliferation and pro-survival gene expression [153,154,155]. Adipose tissue also hosts many immune cells which are significantly altered in the context of obesity [156,157]. 

Given the complex differences in the biology surrounding a cancer developing in the context of obesity, it is likely that the genetic drivers differ in these cancers. Transposon mutagenesis offers an opportunity to reflect these changes in how we model cancer in the mouse. For example, Tschida et al. used *SB* insertion mutagenesis to model hepatocellular carcinoma in the context of steatosis or accumulation of fat in the liver [27]. By comparing steatosis-associated drivers to drivers found in another screen with normal diet [65], the authors were able to identify steatosis-specific drivers. Many published screens should be repeated with the addition of diet-induced obesity in the mice and compared to available normal diet studies to identify targets specific to obesity (Figure 2C). While metastatic driver and therapy resistance studies can be done faithfully with orthotopically implanted tumors, we recommend diet-induced obesity screens be done with the full transposon mutagenesis process. While this is a larger undertaking, it will identify drivers of initiation and early progression in tumor formation, rather than just later drivers of progression and metastasis. These obesity-specific therapies are necessary to address the clearly unmet and increasing need for patients.

## 8. Conclusions and Future Directions

Transposon insertion mutagenesis is a powerful tool to facilitate accelerated evolution and further our understanding of the many dimensions of cancer development, progression, and response to therapy. In this review, we have covered some of the contributions made to the field of cancer biology. These have included models of many types of human cancers, providing insight into the genetic drivers of these cancers as well as powerful pre-clinical model systems. In these studies, the effects of increasing differentiation/linage commitment on cancer development have been studied, as have mechanisms of therapy resistance and metastasis. It was determined that the differentiation status of the tumor cell of origin affects the number of mutations required for tumor formation and the end-tumor expression patterns in surprising ways. Combination therapies have been proposed based on *SB* screens done in the context of a targeted therapy, such as dual treatment with a PI3K inhibitor and vemurafenib in melanoma and FGFR and MEK/ERK pathway inhibitors for breast cancer. In both osteosarcoma and medulloblastoma, drivers found in metastatic clones varied from those in other metastases as well as the primary tumor, indicating that targeted therapies based on the genetics of the primary tumor or even a single metastatic clone are unlikely to eliminate all metastases present.

In the future, we predict transposon mutagenesis will be used to mirror changes in our population by incorporating changes in nutrition in mouse models. This will allow more precise and appropriate therapy to be delivered to patients suffering from obesity in addition to cancer. In addition, transposon mutagenesis studies may help to suggest changes in cancer therapy by identifying resistance mechanisms to targeted therapies as described above and novel ideas for better drug design, including making drugs poor substrates for specific efflux pumps. We predict that transposon screens will be used to identify metastasis-specific drivers in additional tumor types providing additional treatment options for patients with high-risk disease. Lastly, we propose future screens to study the effects of aging on tumor development. For example, it would be interesting to compare screens with mutagenesis initiated in younger versus older mice (>1-year-old) through the use of tamoxifen-induced Cre. These studies may more accurately reflect the development of some tumor types that mainly occur in adult tissues but are poorly modeled by transposon-mutagenesis timed during embryogenesis or early development.

## Figures and Tables

**Figure 1 ijms-21-01172-f001:**
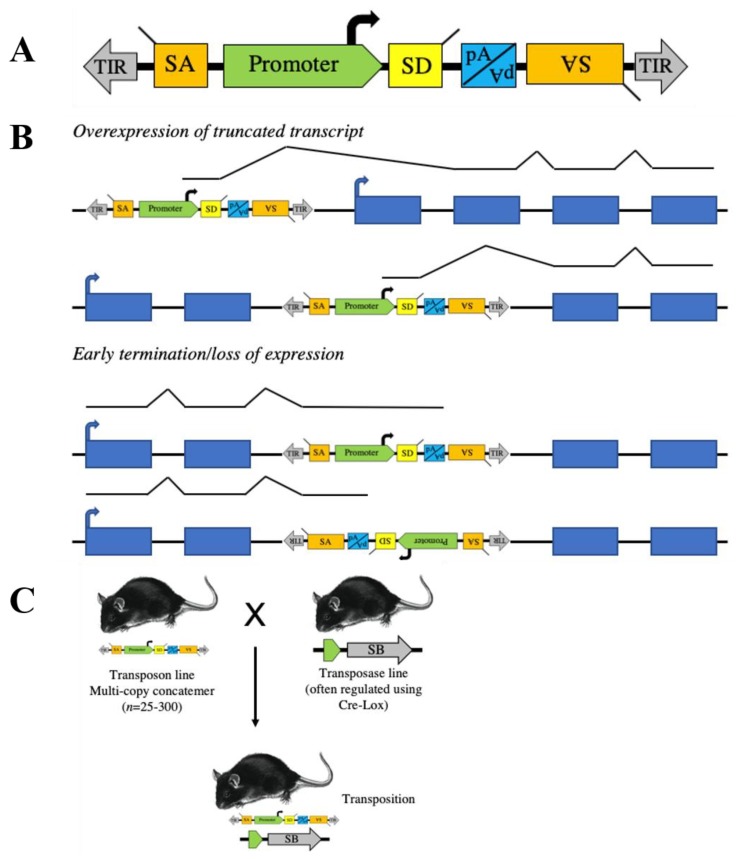
*Sleeping Beauty* (SB) transposons can be designed to randomly induce somatic cell gain and loss of function mutations. (**A**) Structure of a proto-typical transposon vector for somatic cell or cell line mutagenesis studies. A strong promoter followed by an exon with a splice donor (SD) is present to activate transcription of downstream exons. Splice acceptors (SA) and a bi-directional polyadenylation site (pA) are included to disrupt gene expression. (**B**) In mutagenized cells, transposons can activate endogenous proto-oncogenes or disrupt endogenous tumor suppressor genes depending on where insertion occurs and in what orientation. (**C**) Transposon transgenic mice are usually produced by standard pronuclear injection resulting in the generation of lines with multicopy concatomers. These are crossed to mice expressing the transposase to generate mice with somatic cell transposition.

**Figure 2 ijms-21-01172-f002:**
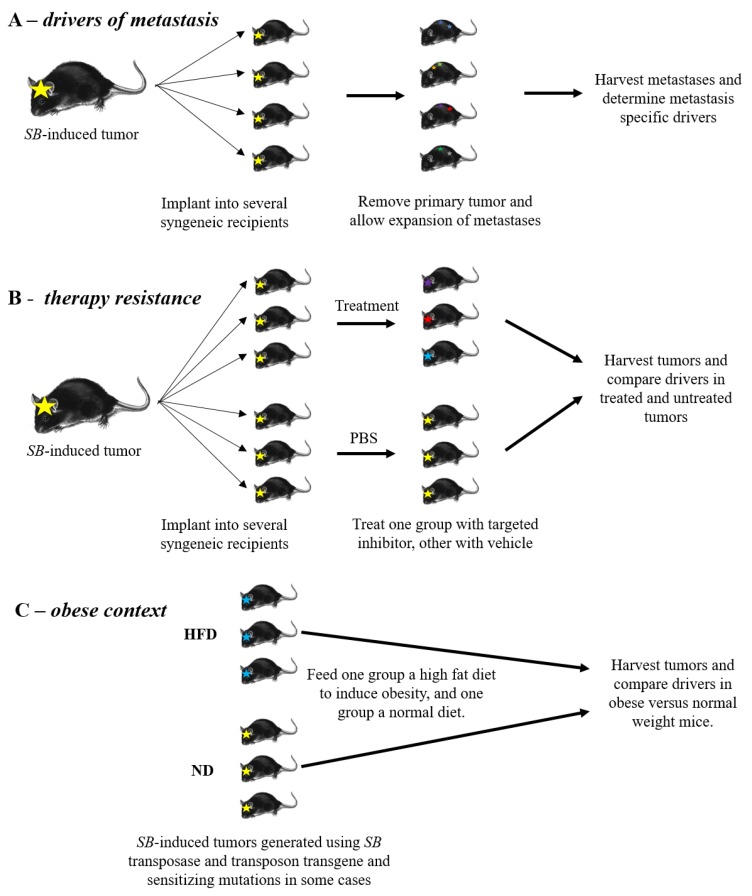
Future experiments for transposon insertional mutagenesis screens. (**A**,**B**) To identify drivers, induce tumor formation with insertional mutagenesis and implant tumors orthotopically into syngeneic recipients. (**A**) The primary tumor is expanded and removed before endpoint, allowing the metastatic lesions to expand and be harvested for driver analysis. (**B**) Implanted tumors are treated with a targeted inhibitor or left untreated and their drivers compared. (**C**) Mice undergoing mutagenesis are fed a normal or high fat diet. At endpoint all tumors are harvested and their drivers compared.

**Table 1 ijms-21-01172-t001:** Systems for Cancer Functional Genomics.

Mutagenesis System	Advantages	Disadvantages
CRISPR/Cas9	Genome wide Useful in loss and gain of function studies Bias can be eliminated by careful guide RNA library design	Difficult to employ in primary cells, useful in only established cell lines Difficult to employ in vivo Difficult to select for phenotypes requiring multiple cooperating genetic alterations
Transposon	Genome wide Useful in loss and gain of function studies Allows screens to be done in cell lines or primary cells in vivo Useful for selection of traits requiring multiple cooperating mutations Non-coding or regulatory regions of the genome can be identified	Bias for or against parts of the genome due to local hopping and insertion site preference Some genes are unlikely to be activated by transposon insertion if first ATG is in exon 1 Some genes are unlikely to be inactivated due to their small size (e.g., microRNAs) Does not induce the full spectrum of mutations found in human cancers (e.g., point mutations and translocations) Transposon mutagenesis can create mutations not tagged by the transposon due to re-mobilization
Retroviruses	Many have been isolated with various tissue tropisms Can activate endogenous promoters by enhancement mechanisms Do not require generation of new transgenic lines of mice	Tend to not induce loss of function mutations, relatively few tumor suppressor genes identified in screens Systems generally found and not created, meaning there are no retroviruses useful for modeling many important types of cancer Tissue tropisms limit usefulness and types of cancer that can be modeled Generally, cells must be dividing for infection Many retroviruses have severe strain-specific effects and limitations

**Table 2 ijms-21-01172-t002:** *Published Sleeping Beauty* and *piggyBac* Cancer Screens in Mice.

Tumor Type	Transposase	Transposon	Cre	Sensitizing Mutations	Refs
*Sarcomas*					
Fibrosarcoma	*CGS-SB10*	*T2/Onc*	-	*p19arf*	[20]
Osteosarcoma	*R26-lsl-SB11*	*T2/Onc*	*Osx-Cre*	*Trp53*	[48]
Peripheral nerve sheath tumor	*R26-lsl-SB11*	*T2/Onc, T2/Onc15*	*Cnp-Cre, Dhh-Cre*	*Trp53, EGFR* *Nf1*	[29,49]
Histiocytic sarcoma	*R26-lsl-SB11*	*T2/Onc, T2/Onc2*	*Lyz2-Cre*	-	[50]
*Carcinomas*					
Skin	*K5-SB11*	*T2/Onc2*	-	*Hras*	[25]
Mammary	*K5-SB11, R26-lsl-SB11*	*T2/Onc2, T2/Onc3*	*K5-Cre, Wap-Cre*	*Trp53, β-catenin, Cdh1, FGFR, Pten*	[51,52,53,54,55]
Pancreatic	*R26-lsl-SB11, R26-lsl-SB13, R26-lsl-PB*	*T2/Onc, T2/Onc2, T2Onc3, ATP1*	*Pdx1-Cre*	*Kras*	[24,56,57]
Gastric adenoma	*R26-lsl-SB11*	*T2/Onc3*	*β-actin-Cre*	*Smad4*	[58]
Intestinal tract	*R26-lsl-SB11*	*T2/Onc, T2/Onc2*	*Vil-CreERT2, Vil-Cre, Ah-Cre*	*Apc, Kras, Smad4, Trp53, Tgfbr2*	[59,60,61,62,63]
Liver	*R26-lsl-SB11*	*T2/Onc, T2/Onc2, T2/Onc3*	*Alb-Cre*	*HBsAg, Trp53, Myc, Steatosis, Pten, Sav1, Met*	[27,64,65,66,67,68,69]
Lung	*R26-lsl-SB11*	*T2/Onc*	*Spc-Cre*	*Trp53, p19arf, Pten*	[70]
Prostate	*CGS-SB10, R26-SB11, R26-lsl-SB11*	*T2/Onc, T2/Onc3*	*PB-Cre*	*p19arf, Pten*	[71,72]
Thyroid	*R26-lsl-SB11*	*T2/Onc2*	*Tpo-Cre*	*Hras*	[73]
*Melanoma*	*R26-lsl-SB11, R26-lsl-SB13, Act-PBase*	*T2/Onc, T2/Onc2, T2/Onc3* *Luc-PB[mut]7*	*Tyr-Cre-ERT2*	*Braf*	[74,75,76,77]
*Hematopoietic*					
T cell leukemia	*R26-lsl-SB11, R26-SB11*	*T2/Onc2*	*Vav-iCre, Lck-Cre, CD4-Cre*	-	[21,78]
T cell lymphoma	*R26-SB11, R26-lsl-PB*	*T2/Onc, ATP2*	*CD4-Cre*	*Trp53, ITK-SYK, Pdc1*	[79,80]
B cell leukemia	*R26-lsl-SB11, Etv6-RUNX1-HSB5*	*T2/Onc*	*Cd79a-Cre*	*Stat5b, Etv6-RUNX1 fusion*	[81,82]
B cell lymphoma	*R26-lsl-SB11, Etv6-RUNX1-HSB5, R26-PB*	*T2/Onc, T2/Onc2, T2/Onc15, ITP1, ITP2*	*Aid-Cre, CD19-Cre, Cnp-Cre*	*Eμ-TCL1, Pax5, Etv6-RUNX1 fusion, Trp53, Pten, Blm*	[23,83,84,85,86]
Acute myeloid leukemia	*R26-lsl-SB11*	*T2/Onc2, GrOnc*	*β-actin-Cre, Vav-Cre, Mx1-Cre*	*Trp53, Jak2, Npm1c, BCR-ABL*	[87,88,89,90]
Mixture of T cell and B cell lymphoma, myeloid leukemia	*R26-SB11*	*T2/Onc*	*-*	*Rassf1a, Cadm1*	[91,92]
Erythroleukemia	*R26-lsl-SB11*	*T2/Onc2*	*Mx1-Cre*	*Cyclin E*	[93]
Myeloid and lymphoid malignancies, thymus, spleen	*R26-lsl-SB11*	*T2/Onc2*	*Vec-Cre*	-	[94]
*Brain tumors*					
Medulloblastoma/CNS-ET *	*R26-lsl-SB11, R26-SB11, Math1-SB11*	*T2/Onc, T2/Onc2, T2/Onc3*	*β-actin-Cre, Nestin-Cre*	*Ptch1, Trp53, Pten*	[28,95,96,97,98,99]
Glioma	*R26-lsl-SB11, R26-SB11*	*T2/Onc, T2/Onc2, T2/Onc3, T2OncATG*	*Nestin-Cre*	*Trp53, p19Arf, Blm, Csf1*	[26,100,101]
*Multiple tumor types*					
Skin, brain, airway, liver, leukemia, lymphoma, intestine	*R26-SB11*	*T2/Onc3*	*-*	*Rag2*	[102]
Leukemia, medulloblastoma, glioma	*CGS-SB10, R26-SB11*	*T2/Onc*	*-*	*p19Arf*	[22]
Skin, liver, lung, brain, lymphoma, sarcoma, mammary, colon, etc.	*R26-SB11*	*T2/Onc3*	-	-	[23]
T cell and B cell leukemia, lymphoma, skin, sarcoma, intestinal tract, lung, liver, etc.	*R26-PB*	*ATP1, ATP2, ATP3*	-	-	[103]
Prostate, mammary and skin carcinomas	*R26-SB11*	*ITP2m*	*-*	*Pten, Blm*	[104]
Sarcoma, carcinoma, leukemia, resistance to MDM2 inhibition	*R26-PB*	*ATP2*	*-*	*p19Arf*	[105]
Liver, lung carcinoma, skin carcinoma, lymphoma	*R26-PB*	*ATP1*	-	-	[106]

* CNS-ET—Embryonal tumor of the central nervous system.

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
