# Peer review of "Transposon Insertion Mutagenesis in Mice for Modeling Human Cancers: Critical Insights Gained and New Opportunities"

_ijms, 2020, doi:10.3390/ijms21031172_

Round 1

Reviewer 1 Report

The manuscript concerns a review that sheds light on the associations of transposon insertion mutagenesis and human cancers in mice model. The review is excellent, and showed a very interesting overview of the role of transposon in cancers. Here are some suggestions that the authors should consider in order to improve the paper.

Major comments:

The actions of sleeping beauty transposon system should be summarized in Figure. The advantages and disadvantages of transposon system should be described and shown in Table or Figure. It is suggested that the authors also add another column describing whether the transposon is related to which of the signaling pathway and function. In Figure 1, please define “Tx” and describe the “outcomes” of those models. It is suggested that the authors should add another table to summarize “driver of metastasis”, “therapy resistance”, “obesity” and “tumor development”. Conclusions and future directions are too short. Please add more comments on this section and indicate the possibilities of the practical use of the results. 

Reviewer 2 Report

The authors present a concise review on the use of transposon-mediated forward genetic studies for discovering new genetic drivers of cancer. They discuss how this approach has been refined to identify cell-type specific initiation events, promoters of metastasis, and factors driving resistance to specific chemotherapeutic drugs. Finally they discuss a relatively recent development in this screening approach, conducting these same screens in the context of diet-induced obesity.  

The review is well written and structured. The senior author is a well-established authority in the review topic.

Major suggested change:

An important variable missing entirely from the review is discussion of gender (and perhaps, aging). Obesity is obviously important and increasingly prevalent, but gender (and aging) affects everyone. These are major factors in cancer biology and are important (controlled?) variables existing in every forward genetic screen.

Minor suggested changes:

The first paragraph introduced transposons as naturally occurring elements but then proceeds to discuss their experimental application with little transition, leading to ambiguities. For example, does the statement “The transposon and transposase can be physically separated…” refer to the former or latter? I would suggest dividing the biology of transposons and the technology of recombinant transposons in different paragraphs.

Line 155 …clinic[al] trials…

Line 216 unnecessary use of hyphen (more-intelligent)

Line 225 …selective pressures of therapy [can] be usefully explored…

Line 238 …the mechanisms behind this are still being understood [?]

Reviewer 3 Report

This is a very good paper with excellent and clear language and logic.

I have only a few comments.

"Transposon insertional mutagenesis" sounds a bit odd to my ear, as "transposon insertion mutagenesis" is the most commonly used term. In fact, we used in one of our papers this "transposon insertional mutagenesis" in the title, but it was changed by the editor to be "transposon insertion mutagenesis". Also "insertional transposon mutagenesis" is often used. This is of course a bit of a matter of taste question, but to consider.

line 33. ...are flanked by terminal inverted repeats.. is not correct. Transposons are typically flanked by a target site duplication. TIRs are part of transposons; located in their termini as the name implies.

line 38. Maybe "Transposition technology" or even "DNA transposition technology" would be better than "transposon technology"

line 48. This sentence states that transposons have been used succesfully to study gene function in several eukaryotes.

Somehow the phrasing implies that this is so in only those organisms mentioned. So better modify this beginning by stating the usage in a more broad sense. Especially as such endeavours have been highly successfully initiated with bacteria and e.g. with yeast. Nowadays also in archaea (e.g. papers dealing with the archaeal model organism Haloferax volcanii, Kiljunen et al 2014 BMC Biol; Legerme et al 2016 Life; Gomez et al 2018 Genes). 

lines 48-81; i.e. the two chapters: the chapter describes quite well the systems used and their pros and cons. However, all these are of eukaryotic origin.  Importantly, it has been shown that a transposon originated from a prokaryotic system can also be used to modify eukaryotic cells efficiently (delivery of phage Mu transpososomes, Paatero et al 2008 Nucleic Acids Res). It would be good to add this (and possibly other prokaryotic systems functioning in eukaryotes - if somewhere described (altough I am currently not aware of such systems)). The  idea is that we are  not limited only to the use of originally eukaryotic transposons. Especially as transposases can be mutagenized  for better performance relatively easily (see e.g. Rasila et 2018 Nucleic Acids Res), prokaryotic transposons  bear considerable potential also in eukaryotes, especially because  of their abundance and diversity.       

Reviewer 4 Report

This is a highly educational review article from the Largaespada lab describing the use of transposon mutagenesis for the discovery of novel cancer genes in mouse models. I specifically liked the sections proposing to include a high fat diet in cohorts of experimental animals to model obesity in the context of cancer. I have a couple of suggestions for additions into the paper that I believe readers would appreciate.

1) Authors describe a mutagenic transposon vector that can simultaneously trigger gain of function and loss of function mutations in genes, driven by splice sites, promoters and polyA signals inside the mutagenic transposon. I understand the argument that such arrangement can cover a wide array of genotypes (coverage is large), but at the same time it complicates downstream molecular analyses, because it is more difficult to tell what molecular events led to phenotypes. It would be useful to have some discussion on the relative advantages/disadvantages of coupling such mutagenic signals inside the same transposon as opposed to crafting vectors that would only do gain of function and vectors that would only do loss of function, and using them separately.

2) The reader can appreciate the power of transposon mutagenesis, but other technologies are also available to deregulate gene expression in phenotypic screens. For example, CRISPRi and CRISPRa screens have been shown to be highly useful to generate phenotypes, at least ex vivo. It would be highly appreciated if authors could critically review and compare these technologies with respect to their relative advantages and disadvantages.

3) page 5, lines 178-182. The proposed scheme for screening for metastatic drivers is elegant, but I wonder if there's enough time for i) the evolution of metastatic tumors in vivo, given the burden of a transplanted primary tumor in the same animal (animals may simply die from the primary tumor before they develop metastatic tumors) or ii) the development of metastatic tumors in vivo if primary tumors are removed too early. In other words, such screens may prove tricky with respect to settling for a time window that is necessary to keep animals alive but at the same time necessary for the formation of metastasis. It would be helpful to discuss this.

4) page 2, line 64. Authors cite papers for the extended cargo capacity of the PB transposons. There are other publications out there, including Rostovskaya et al 2012 NAR and Wang et al 2014 NAR, that show that PB's cargo capacity is not superior to SB's cargo capacity. For a balanced presentation of the literature, authors should include those references with a brief discussion.

5) page 7, line 238. "... the mechanisms behind this are still being understood." sounds a bit awkward. Maybe "... the mechanisms behind this are still being investigated" or "... the mechanisms behind this are still incompletely understood" would be more appropriate.

Round 2

Reviewer 1 Report

The authors have adequately responded to most of my previous concerns.